# Crumbs and the apical spectrin cytoskeleton regulate R8 cell fate in the *Drosophila* eye

**Jonathan M. Pojer**[1,2], **Abdul Jabbar Saiful Hilmi**[1,2], **Shu Kondo**[3], **Kieran F. Harvey**[1,2,4]*

**1** Peter MacCallum Cancer Centre, Melbourne, Victoria, Australia, **2** Sir Peter MacCallum Department of Oncology, The University of Melbourne, Parkville, Victoria, Australia, **3** Laboratory of Invertebrate Genetics, National Institute of Genetics, Mishima, Shizuoka, Japan, **4** Department of Anatomy and Developmental Biology, Monash University, Clayton, Victoria, Australia

* kieran.harvey@petermac.org

## Abstract

The Hippo pathway is an important regulator of organ growth and cell fate. In the R8 photo-receptor cells of the *Drosophila melanogaster* eye, the Hippo pathway controls the fate choice between one of two subtypes that express either the blue light-sensitive Rhodopsin 5 (Hippo inactive R8 subtype) or the green light-sensitive Rhodopsin 6 (Hippo active R8 sub-type). The degree to which the mechanism of Hippo signal transduction and the proteins that mediate it are conserved in organ growth and R8 cell fate choice is currently unclear. Here, we identify Crumbs and the apical spectrin cytoskeleton as regulators of R8 cell fate. By contrast, other proteins that influence Hippo-dependent organ growth, such as the baso-lateral spectrin cytoskeleton and Ajuba, are dispensable for the R8 cell fate choice. Surprisingly, Crumbs promotes the Rhodopsin 5 cell fate, which is driven by Yorkie, rather than the Rhodopsin 6 cell fate, which is driven by Warts and the Hippo pathway, which contrasts with its impact on Hippo activity in organ growth. Furthermore, neither the apical spectrin cyto-skeleton nor Crumbs appear to regulate the Hippo pathway through mechanisms that have been observed in growing organs. Together, these results show that only a subset of Hippo pathway proteins regulate the R8 binary cell fate decision and that aspects of Hippo signal-ling differ between growing organs and post-mitotic R8 cells.

## Author summary

Signalling pathways operate throughout living organisms to allow them to detect different stimuli and control appropriate responses to them. The Hippo pathway is one such signal-ling pathway, which operates in many different organisms to control the ability of cells to proliferate, die and differentiate. The mechanism by which the Hippo pathway signals to control cell proliferation and apoptosis during the growth of different organs has been intensely studied but the mechanism by which it controls cell fate is relatively poorly understood. In the present manuscript, we report the discovery of new insights into how the Hippo pathway communicates to control the fate of specific light-sensing cells (R8 cells) in the *Drosophila* eye and how this differs from Hippo pathway signalling in organ

**Data Availability Statement:** All relevant data are within the manuscript and its Supporting Information files.

**Funding:** K.F.H was supported by the National Health and Medical Research Council of Australia

(1078220 and 1194467) (https://www.nhmrc.gov.
au/). J.M.P. was supported by an Australian
Postgraduate Award (https://scholarships.unimelb.
edu.au/awards/graduate-research-scholarships).
This research was supported by a grant to K.F.H.
from the Australian Research Council
(DP180102044) (https://www.arc.gov.au/). The
funders had no role in study design, data collection
and analysis, decision to publish, or preparation of
the manuscript.

**Competing interests:** The authors have declared
that no competing interests exist.

growth. Our discoveries shed new light on how the eye develops in order to visualize different colours and how a key developmental signalling pathway is redeployed to perform distinct roles.

# Introduction

Binary cell fate decisions allow for the specification of a large number of cell subtypes from a small number of precursor cells. In the nervous system, binary cell fate decisions lead to a diverse range of nearly identical cells that respond to different stimuli [1–3]. One such binary fate choice occurs in the R8 photoreceptor cells of the *Drosophila melanogaster* eye. The adult *D. melanogaster* compound eye is composed of an array of around 800 subunits, called ommatidia, each of which contains eight photoreceptor cells (R1-R8). These cells are defined by a specialised subcellular compartment called the rhabdomere, which is composed of tens of thousands of microvilli that project from the cell body of each photoreceptor into the inter-rhabdomeric space at the centre of each ommatidium. The rhabdomeres of the R7 and R8 photoreceptor cells are arranged in tandem and share the same optic path, with the R7 cell positioned distally and the R8 cell positioned proximally (**Fig 1A and 1A'**) [4]. Each photoreceptor cell expresses a specific rhodopsin, a photosensitive G protein-coupled receptor with a distinct spectral sensitivity [5, 6]. Expression of distinct rhodopsins in different photoreceptor cells allows each cell to respond to specific wavelengths of light and prevent sensory overlap. The outer photoreceptors, R1-6, express *Rh1* and allow *D. melanogaster* to detect motion [7–9], while the inner photoreceptors, R7 and R8, express one of *Rh3*, *Rh4*, *Rh5* or *Rh6*, and are the primary cells that mediate colour vision [10].

There are different subtypes of ommatidia in the *D. melanogaster* eye, which differ based on the rhodopsins expressed in the R7 and R8 cells. The dominant subtypes are known as the 'pale' (**p**) and 'yellow' (**y**) subtypes. The **p** subtype accounts for around 30% of all ommatidia, with the short UV-sensitive *Rh3* being expressed in **p**R7 cells and the blue-sensitive *Rh5* being expressed in **p**R8 cells; the **y** subtype accounts for the remaining ~70% of ommatidia, with the long UV-sensitive *Rh4* being expressed in **y**R7 cells and the green-sensitive *Rh6* being expressed in **y**R8 cells (**Fig 1B**). Specification of the inner photoreceptor cells is linked to ensure that the rhodopsins expressed in each subtype are always matched between R7 and R8 cells. In the late pupal retina, the transcription factor Spineless is expressed stochastically in ~70% of R7 cells, inducing **y**R7 cell fate and *Rh4* expression. The remaining R7 cells take on a **p**R7 cell fate and express *Rh3* [11, 12]. In these cells, the Transforming growth factor-β pathway is activated, signalling to the neighbouring R8 cell to take on a **p**R8 cell fate [13].

R8 cell fate is specified through a bistable feedback loop composed of the kinase Warts (Wts), the transcriptional coactivator Yorkie (Yki), and the Pleckstrin-homology domain protein Melted (Melt) [14, 15] (**Fig 1B**). Yki and Wts are key components of the Hippo pathway, an important regulator of organ growth and cell fate [16–18] (**Fig 1C**). In growing organs, such as the larval imaginal discs, a kinase cassette, composed of the serine/threonine kinases, Hippo (Hpo), a sterile-20-like (Ste20) kinase [19–22], and Wts, a nuclear DBF2-related (NDR) kinase [23–25] and the scaffolding factors, Salvador (Sav) [25, 26] and Mob as tumour suppressor (Mats) [27, 28], inactivate the WW-domain containing transcriptional coactivator, Yki [29] (**Fig 1C'**). Yki cannot bind to DNA itself, so must interact with transcription factors, such as the TEAD/TEF transcription factor, Scalloped (Sd), to regulate expression of target genes [30–32].

Upstream of the core kinase cassette, the Hippo pathway integrates signals from surrounding cells and the extracellular matrix to regulate Yki activity [16, 33–35]. Key upstream

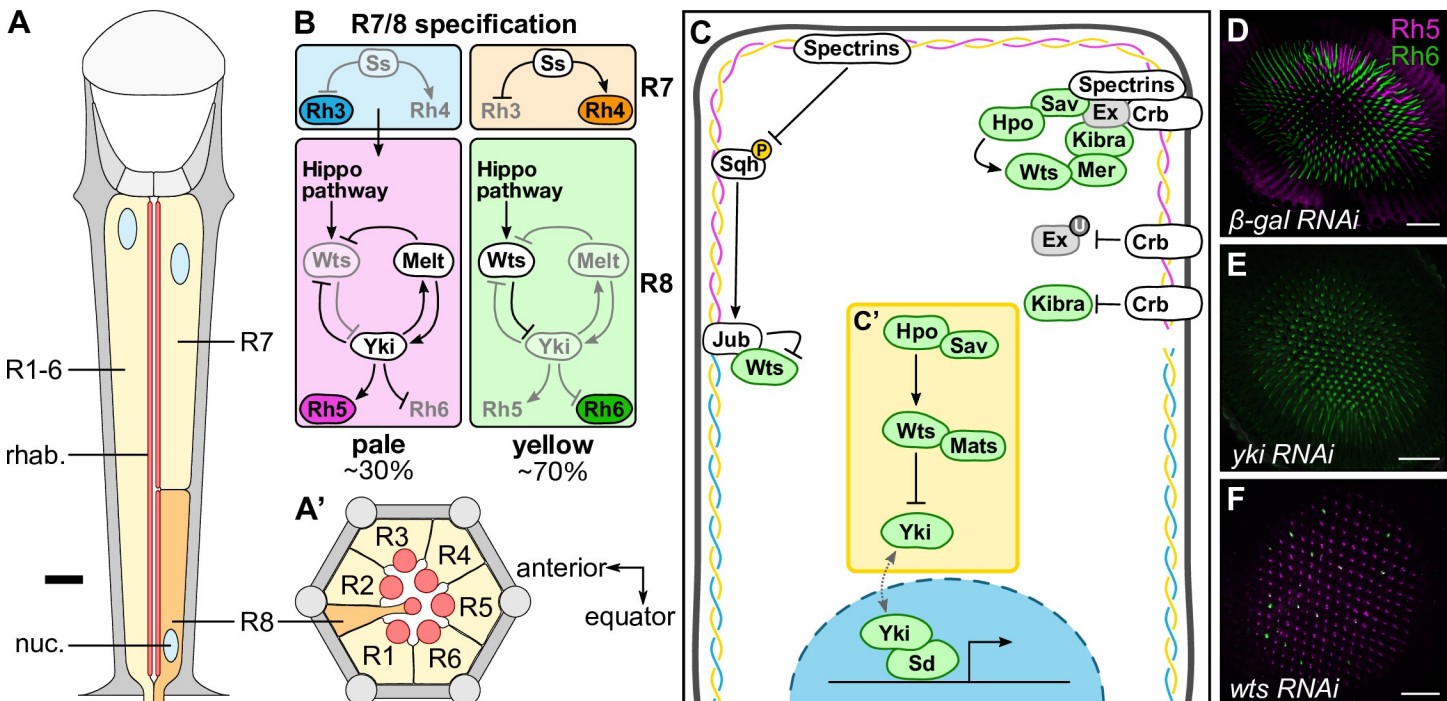

**Fig 1. Regulation of *Drosophila melanogaster* R8 cell fate by the Hippo pathway.** (A-A') Schematic diagram of a *D. melanogaster* ommatidium. Yellow cells are R1-7 photoreceptor cells; orange cells are R8 photoreceptor cells; grey cells are other cells in the ommatidium. Blue circles are photoreceptor nuclei (nuc.); red lines/circles are rhabdomeres (rhab.). (**A**) Longitudinal section of an ommatidium. Note that R7 and R8 cells share the same optic path. The thick black line indicates approximately where the transverse section (**A'**) is drawn from. The distal section of the retina (towards the lens and outer surface of the eye) is to the top; the proximal section of the retina (towards the brain) is to the bottom. (**A'**) Transverse section of the proximal section of an ommatidium, showing the R8 cell. The anterior of the retina is to the left; the equator of the retina is to the bottom. (**B**) The main photoreceptor subtypes, showing R7 and R8 cell specification in each subtype. In the pale subtype, the R7 cell expresses *Rh3* (blue), signalling to the R8 cell to take on a **p**R8 cell fate through a bistable loop composed of Warts (Wts), Melted (Melt) and Yorkie (Yki) and promoting expression of *Rh5* (magenta). In the yellow subtype, the R7 cell expresses Spineless (Ss) which promotes *Rh4* (orange), while the R8 cell expresses *Rh6* (green). The subtypes are found in the specified proportions. (**C-C'**) Schematic of the Hippo pathway in epithelial tissue growth. Proteins labelled in green regulate R8 cell fate; proteins labelled in grey do not regulate R8 cell fate; proteins labelled in white have not been studied in R8 cell fate. The spectrin cytoskeleton is shown beneath the plasma membrane, highlighting the three spectrin proteins: α-Spec (yellow), β-Spec (cyan) and Kst (magenta). The yellow box (**C'**) highlights the core kinase cassette. Crb, Crumbs; Ex, Expanded; Hpo, Hippo; Jub, Ajuba; Mats, Mob as tumour suppressor; Mer, Merlin; Sav, Salvador; Sd, Scalloped; Sqh, Spaghetti squash; Wts, Warts; Yki, Yorkie. (**D-F**) Confocal microscope images of adult *D. melanogaster* retinas stained with anti-Rh5 (magenta) and anti-Rh6 (green) antibodies. The indicated RNAi lines were driven by *lGMR-Gal4*. Retinas expressing *β-gal RNAi* had a wild type ratio of R8 subtypes (**D**); retinas expressing *yki RNAi* had almost exclusively **p**R8 cells (**E**); retinas expressing *wts RNAi* had almost exclusively **y**R8 cells (**F**). Scale bars are 50µm.

regulators of the Hippo pathway that control organ growth include the 4.1/ezrin/radixin/moe-sin (FERM) domain proteins, Merlin (Mer) and Expanded (Ex), and the WW-domain protein, Kibra [36–43]; the Ste20 kinase, Tao [44, 45]; the polarity proteins, Crumbs (Crb), Lethal (2) giant larvae (Lgl), and the atypical cadherins Fat (Ft) and Dachsous (Ds) [42, 46–52]; and mechanosensors, such as the spectrin cytoskeleton and Ajuba (Jub) [53–59] (**Fig 1C**). These proteins are enriched in particular subcellular domains, with many of them, including Crb, the apical spectrin cytoskeleton, Mer, Kibra and Ex, localising to apical membrane domains and sub-apical regions [43, 54, 60].

Many of these upstream Hippo pathway proteins also control the fate of R8 cells, which are post-mitotic. Upstream Hippo pathway proteins, such as Mer, Kibra and Lgl, converge on the core Hippo pathway kinases, Hpo and Wts in R8 cells, as in organ growth. Active Wts in **y**R8 cells prevents Yki from promoting the **p**R8 cell fate, and allows *Rh6* to be expressed [15]. Conversely, in **p**R8 cells, Wts is inactive, allowing Yki to bind to Sd and directly promote transcription of *Rh5* [15]. Yki is involved in two feedback loops in R8 cells: (1) a positive feedback loop, where Yki promotes transcription of *melt*, promoting its own activation; and (2) a double-

negative feedback loop, where Yki represses transcription of *wts*, thereby preventing its own repressor from acting on it [15, 61] (**Fig 1B and 1D–1F**). This bistable feedback loop ensures that only one type of rhodopsin is expressed in each R8 cell.

Other Hippo pathway proteins, such as Ex, Ft and Ds are not required for the R8 cell fate choice [62], suggesting there are differences in how the Hippo pathway functions in different biological settings. Currently, however, we lack a complete understanding of which Hippo pathway proteins control R8 cell fate and how upstream regulators control the Hippo pathway in these cells. Here, we investigated the spectrin cytoskeleton, Crb and Jub in R8 cell fate. We identified α-Spec and Kst, components of the apical spectrin cytoskeleton, as promoters of **y**R8 cell fate and Crb as a promoter **p**R8 cell fate. By contrast, β-Spec and Jub were found to not play a role in R8 cell fate specification. Furthermore, the apical spectrin cytoskeleton and Crb appear to regulate the Hippo pathway in post-mitotic R8 cells in manners distinct from how they function in actively growing organs.

## Results

### The apical spectrin cytoskeleton promotes yR8 cell fate

To better understand Hippo signalling in R8 cells, we performed a systematic search of Hippo pathway proteins that have been implicated in organ growth but not R8 cell fate control. To assess potential roles for these proteins in R8 cell fate regulation, we used mutant alleles or genetically depleted components of the Hippo pathway in all photoreceptors using published RNAi lines and the *long Glass Multiple Reporter* (*lGMR*)-*Gal4* driver [63]. The ratio of R8 sub-types was determined by assessing the number of R8 cells that expressed Rh5 or Rh6, relative to control eyes (**Fig 1D**, *β-gal RNAi*, approximately 30% **p**R8 cells, consistent with previous studies [14, 62]). Through this screen (to be described elsewhere), we identified roles for the apicobasal polarity protein Crb and the apical spectrin cytoskeleton in the control of R8 cell fate.

The spectrin cytoskeleton is a network of large proteins that form on the intracellular surface of the plasma membrane and is widely conserved in animals [64]. In *D. melanogaster*, it is composed of tetramers of α-Spectrin (α-Spec) and one of the two β-spectrin homologues, β-Spectrin (β-Spec) or Karst (Kst, also $β_{Heavy}$-Spectrin). These tetramers are spatially distinct in epithelial cells, with α-β tetramers localising to the basolateral membrane, and α-Kst tetramers at the apical membrane [65]. The spectrin cytoskeleton has been reported to regulate Hippo pathway activity both by responding to mechanical forces and by regulating the accumulation of upstream Hippo pathway proteins at specific plasma membrane domains [54, 57]. Both spectrin cytoskeleton forms regulate the Hippo pathway in *D. melanogaster*, although differences have been reported in which of these operate in different tissues [53–56].

To investigate the role of the spectrin cytoskeleton in R8 cell fate, we depleted each spectrin gene using RNAi. Depletion of *α-Spec* (approximately 60% **p**R8 cells, p<0.0001; approximately 40% **p**R8 cells, p = ns) and *kst* (43–75% **p**R8 cells across two RNAi lines, p<0.0010) in photoreceptor cells resulted in an increase in the proportion of **p**R8 cells, while depletion of *β-Spec* did not change the ratio of R8 subtypes (24–31% **p**R8 cells across three RNAi lines, p = ns) (**Figs 2A–2D and S1A–S1D**). This suggests that the apical, but not the basolateral, spectrin cytoskeleton regulates R8 cell fate. This is consistent with temporally-distinct roles of the different spectrin cytoskeletons in pupal eye development, where the basolateral spectrin cytoskeleton is required for photoreceptor morphogenesis in the mid-pupal eye, while the apical spectrin cytoskeleton is required for photoreceptor morphogenesis in the late pupal eye, which coincides with when R8 subtypes are specified [66, 67].

To confirm that the apical spectrin cytoskeleton regulates R8 cell fate through the Hippo pathway, we depleted *yki* in addition to *α-Spec* or *kst*. As expected, in these scenarios the

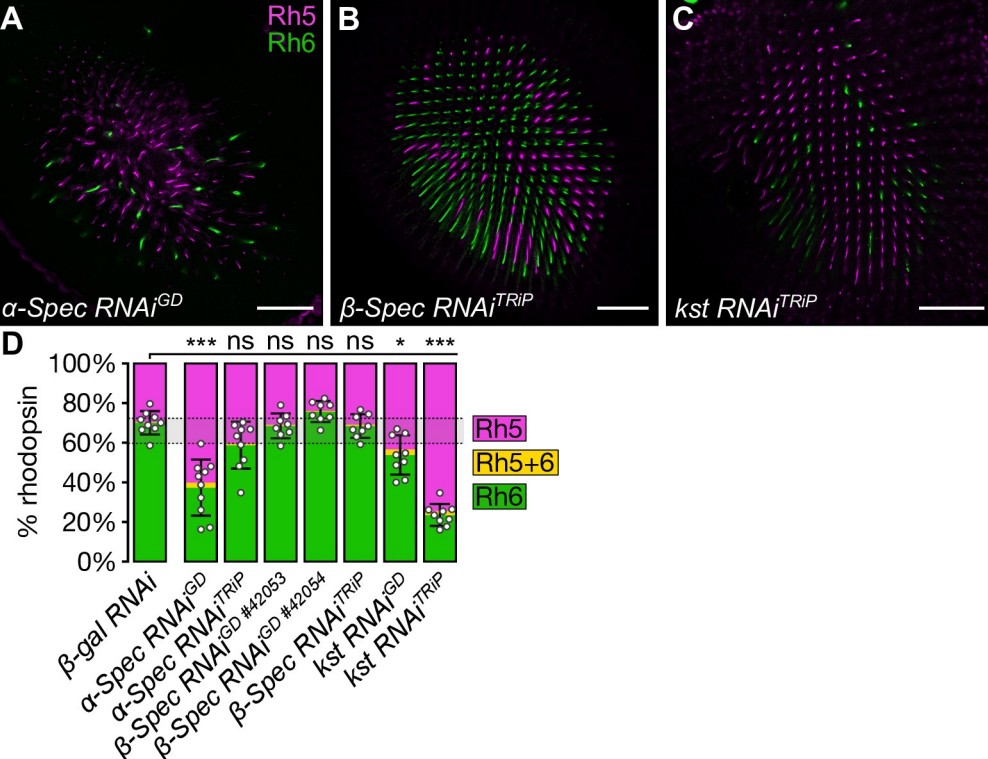

**Fig 2. The apical spectrin cytoskeleton promotes pR8 cell fate.** (**A-C**) Confocal microscope images of adult *D. melanogaster* retinas stained with anti-Rh5 (magenta) and anti-Rh6 (green) antibodies. The indicated RNAi lines were driven by *lGMR-Gal4*. Retinas expressed *α-Spec RNAi^{GD}* (**A**), *β-Spec RNAi^{TRiP}* (**B**) and *kst RNAi^{TRiP}* (**C**). Scale bars are 50μm. (**D**) Proportion of R8 cells that express Rh5 (magenta), Rh6 (green), or both (yellow). The error bars represent the standard deviation of total % Rh5 (% cells expressing only Rh5 and cells co-expressing Rh5 and Rh6). Total % Rh5 was compared with two-sided, unpaired t-tests; ns = not significant, $^* = p<0.01$, $^{***} = p<0.0001$. The shaded grey region between the dotted grey lines indicates the wild type Rh5:Rh6 ratio range. *β-gal RNAi* (**Fig 1D**): n = 9 retinas, 3976 ommatidia; *α-spec RNAi^{GD}*: n = 8, 1466; *α-spec RNAi^{TRiP}* (**S1A Fig**): n = 9, 2207; *β-spec RNAi^{GD #42053}* (**S1B Fig**): n = 8, 2882; *β-spec RNAi^{GD #42054}* (**S1C Fig**): n = 8, 2188; *β-spec RNAi^{TRiP}*: n = 8, 2776; *kst RNAi^{GD}* (**S1D Fig**): n = 9, 3689; *kst RNAi^{TRiP}*: n = 9, 2361.

majority of R8 cells expressed Rh6 (**S4 Fig**), suggesting that α-Spec and Kst act upstream of Yki. Similarly, depleting *α-Spec* or *kst* in conjunction with overexpression of the upstream Hippo pathway protein *kibra*, which functions in parallel to the apical spectrin cytoskeleton in larval imaginal discs [54], resulted in almost all R8 cells expressing Rh6 (**S4 Fig**), suggesting that the apical spectrin cytoskeleton regulates R8 cell fate upstream of, or in parallel to, Kibra. Finally, we investigated the expression of a reporter of *wts* transcription, *wts-LacZ*, given that like *Rh5*, *wts* is a direct target gene of Yki and Sd in R8 cells. Upon depletion of components of the apical spectrin cytoskeleton, *wts-LacZ* was still confined to Rh6+ yR8 cells (**S4 Fig**), indicating that the apical spectrin cytoskeleton does not regulate R8 cell fate independently of *wts* transcription. Collectively, these results are consistent with the notion that the apical spectrin cytoskeleton modulates R8 cell fate through Hippo-mediated transcription.

## The subcellular localisation of α-Spectrin in R8 cells changes during late pupal eye development

Subcellular localisation is essential both for proper function of the spectrin cytoskeleton, as well as the Hippo pathway. While the two *D. melanogaster* β-Spectrin proteins, β-Spec and Kst

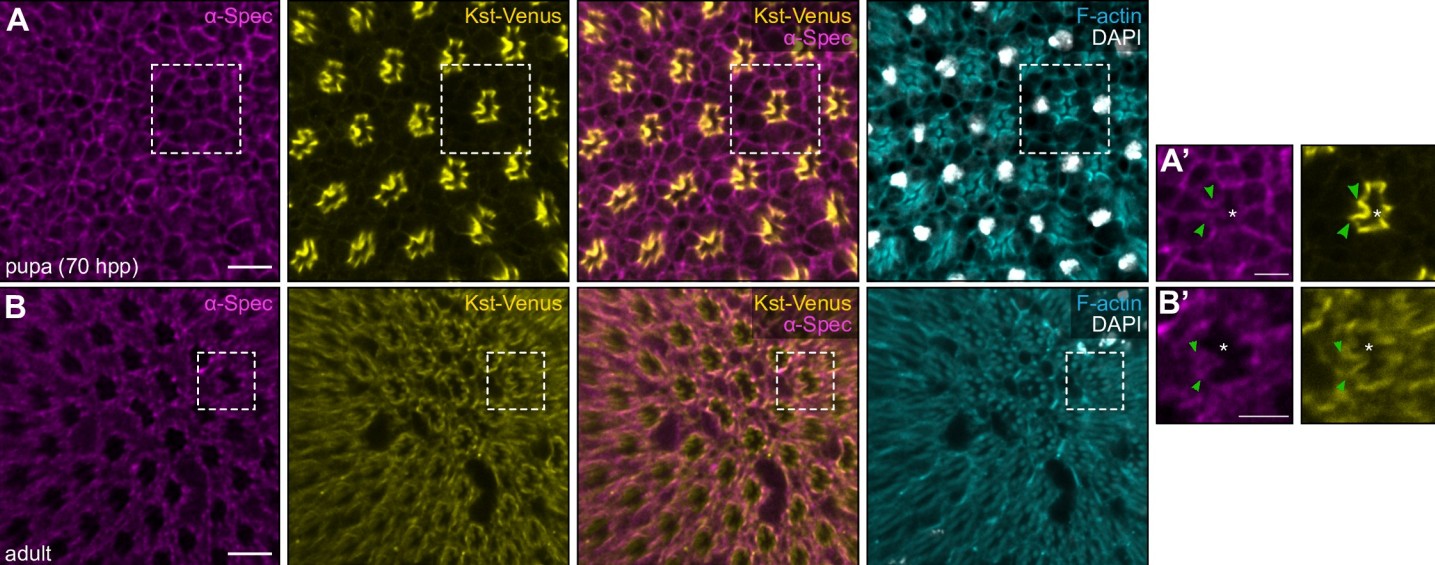

**Fig 3. Subcellular localisation of α-Spectrin differs between late pupal and adult photoreceptor cells.** (**A-B'**) Confocal microscope images of late pupal (70 hours after pupariation formation, APF) and adult *D. melanogaster* retinas. Endogenously tagged Kst-Venus retinas were stained with an anti-GFP antibody to amplify the Venus signal, an anti-α-Spec antibody, DAPI (white; nuclei) and Rhodamine Phalloidin (cyan; F-actin in rhabdomeres and cell membranes). In each image, anterior is to the left. The dashed white boxes in **A** and **B** indicate the area shown in **A'** and **B'**, respectively. White asterisks indicate the rhabdomere of the ommatidium; green arrowheads indicate the adherens junctions. Scale bars are 10μm in **A** and **B**; and 5μm in **A'** and **B'**.

localise at the basolateral and apical membranes, respectively, localisation of α-Spec can vary depending on which spectrin complex it forms. In the photoreceptor precursors in the larval imaginal eye disc, α-Spec localises at the apical domain, while in photoreceptors in mid-pupal eyes α-Spec localises primarily at the basolateral membrane domains and more weakly at the apical domains [66]. While the spectrins do not play an obvious role in early photoreceptor differentiation, the basal enrichment of α-Spec during the mid-pupal stage of development corresponds with an increased dependency on the basolateral spectrin cytoskeleton for morphogenesis [66]. As the Hippo pathway is important for both the specification of R8 cell fate in late pupal retinas, and the maintenance of R8 cells fate in adult eyes [62], we investigated the localisation of the apical spectrin cytoskeleton components at both stages of development. Surprisingly, 70 hours after pupariation formation (APF), when R8 cells begin to be specified [62], α-Spec predominantly localised at the basal membrane of R8 cells, while endogenously tagged Kst (Kst-Venus) (**S7A Fig**) localised exclusively at the apical membrane (**Fig 3A and 3A'**). This result is reminiscent of the localisation of the spectrins in photoreceptor cells during mid-pupal development, when α-Spec and β-Spec control photoreceptor morphogenesis and Kst is dispensable for this process [66]. Conversely, in adult R8 cells, which rely on the Hippo pathway to maintain their fate [62], we found that both α-Spec and Kst-Venus colocalised at the apical membrane (**Fig 3B and 3B'**). This suggests that there is a switch of the dominant spectrin cytoskeleton form in R8 cells between late pupae and adults.

## The apical spectrin cytoskeleton influences R8 cell fate independent of Spaghetti squash activity

Two models have been proposed to explain how the apical spectrin cytoskeleton influences Hippo pathway activity: (1) it influences the phosphorylation and activation of the regulatory light chain of myosin II, Spaghetti squash (Sqh), and thereby modulates cortical tension–upon

spectrin loss, cortical tension increases at adherens junctions leading to increased Jub-dependent tethering of Wts and therefore reduced Wts activity and elevated Yki activity [53, 56, 57]; and (2) spectrins recruit core Hippo pathway proteins to the sub-apical regions through the apicobasal polarity protein Crb–upon spectrin loss, Hippo activation complexes are depleted and Wts activity is reduced, causing Yki hyperactivation [54] (**Fig 1C**). Consistent with published studies [57], upon depletion of either *α-Spec* or *kst*, phosphorylated Sqh (pSqh) was increased in photoreceptor cells (**Fig 4A–4D**), however total Sqh levels remained unaffected (**S6 Fig**). To determine whether this change in levels of pSqh could account for the change in R8 subtypes seen upon depletion of the apical spectrin cytoskeleton, we misexpressed constitutively active forms of Sqh (*sqh.EE*), and Rho-associated kinase (Rok; *rok.CA*), which phosphorylates Sqh [68]. While misexpression of *rok.CA* did not change the R8 subtype ratio (around 34% **p**R8 cells), misexpression of *sqh.EE* resulted in a weak, but significant, increase in the proportion of **y**R8 cells (around 24%, p<0.0001) (**S6 Fig**). Surprisingly, this is the opposite phenotype to that observed upon depletion of the apical spectrin cytoskeleton, suggesting that increased pSqh does not drive Yki hyperactivation and pR8 cell fate following apical spectrin cytoskeleton disruption.

To further investigate the role Sqh on the Hippo pathway in R8 cells, we assessed the expression and subcellular localisation of a Jub-GFP transgene [69] (**S2 Fig**), given that changes in Sqh activity can modulate Jub recruitment to adherens junctions in growing imaginal discs [59]. While Jub-GFP colocalised with E-Cadherin at adherens junctions in larval eye imaginal discs, we could not detect Jub-GFP expression in adult photoreceptor cells (**Fig 4E and 4F**). To further interrogate this, we investigated a potential role for *jub* in R8 cell fate and found that expression of *jub* RNAi lines did not alter the R8 subtype ratio (21–30% **p**R8 cells across two RNAi lines, p = ns) (**Figs 4G and 4H and S1E**). This suggests that while the apical spectrin cytoskeleton influences Sqh activity in R8 cells, this does not mediate its impact on the Hippo pathway in these cells.

### Crumbs promotes pR8 cell fate

An alternative mechanism by which the apical spectrin cytoskeleton has been proposed to regulate the Hippo pathway is by forming a complex with Crb at sub-apical regions [54]. Kst and Crb physically interact in a number of *D. melanogaster* tissues, including embryos and pupal photoreceptors, where they promote correct apical domain formation [66, 67, 70]. In larval wing imaginal discs, Crb and Kst colocalise at the sub-apical region and have been reported to promote accumulation of several Hippo pathway proteins at these junctions, including Ex, Mer, Kib, Hpo and Wts, leading to Hippo pathway activation and suppression of Yki-mediated tissue growth [54]. As Crb and Kst colocalise in pupal and adult photoreceptor cells at the stalk membrane (the apical membrane below the rhabdomere [66, 67]), we hypothesised that they also recruit Hippo pathway proteins to the stalk membrane and promote its activation in this membrane domain. To investigate a role for Crb in R8 cell fate, we used *eyFlp/FRT* site-specific recombination [71] to generate clones of tissue harbouring the *crb* null allele, *crb^{11A22}* [72] (**Figs 5A, 5B and 5D and S2A and S2C**). *crb^{11A22}* clones displayed two distinct phenotypes that distinguished them from wild type clones. First, *crb^{11A22}* clones had a reduction in the proportion of **p**R8 cells by around 3.6 times when compared with neighbouring wild type clones (**Figs 5A and 5D and S2C**). This was surprising, as it suggests that Crb actually promotes **p**R8 cell fate, while the apical spectrin cytoskeleton and other Hippo pathway proteins with which Crb has been reported to function within the context of epithelial tissue growth, such as Wts, Hpo and Sav, promote the opposing **y**R8 cell fate. Second, the rhabdomeres of photoreceptors in *crb^{11A22}* clones were shortened, suggesting a fault in photoreceptor

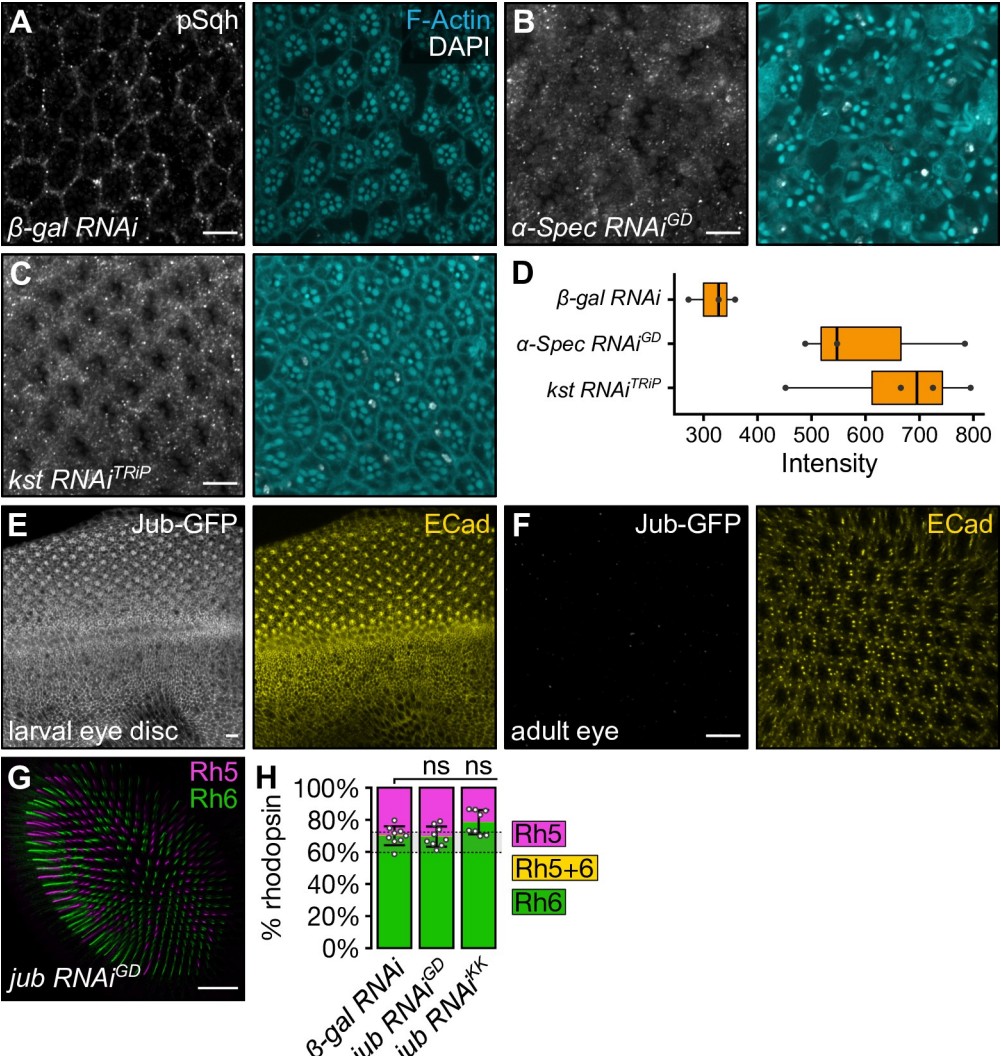

**Fig 4. The apical spectrin cytoskeleton modulates phosphorylation of Spaghetti squash.** (**A-C**) Confocal microscope images of adult *D. melanogaster* eyes stained with anti-pSqh (grey) antibody, DAPI (white) and Phalloidin (F-actin, cyan). The indicated RNAi lines were driven by *lGMR-Gal4*. Retinas expressed *β-gal RNAi* (**A**), *α-Spec RNAi*$^{GD}$ (**B**), or *kst RNAi*$^{TRiP}$ (**C**). Scale bars are 10μm. (**D**) Boxplot showing intensity of pSqh in (**A-C**). (**E-F**) Confocal microscope images of a *Jub-GFP D. melanogaster* larval eye imaginal disc (**E**) and an adult eye (**F**). Tissues were stained with anti-GFP (grey) and anti-ECad (yellow) antibodies. Scale bars are 10μm. (**G**) Confocal microscope images of adult *D. melanogaster* retina stained with anti-Rh5 (magenta) and anti-Rh6 (green) antibodies. Expression of *jub RNAi*$^{GD}$ was driven by *lGMR-Gal4*. Scale bar is 20μm. (**H**) Proportion of R8 cells that express Rh5 (magenta), Rh6 (green), or both (yellow). The error bars represent the standard deviation of total % Rh5 (% cells expressing only Rh5 and cells co-expressing Rh5 and Rh6). Total % Rh5 was compared with a two-sided, unpaired t-test; ns = not significant. The shaded grey region between the dotted grey lines indicates wild type Rh5:Rh6 ratio range. *β-gal RNAi* (**Fig 1D**): n = 9 retinas, 3976 ommatidia; *jub RNAi*$^{GD}$: n = 9, 3660; *jub RNAi*$^{KK}$ (**S1E Fig**): n = 8, 3048.

morphogenesis (**S3A Fig**), a phenotype that has been previously described as a failure of rhabdomere elongation in pupal development [73].

## Crumbs regulates R8 cell fate through its FERM-binding motif

A question that arose from these observations was whether these two phenotypes–the change in R8 subtype ratio and the disruption of rhabdomere morphogenesis–were linked. Crb is a

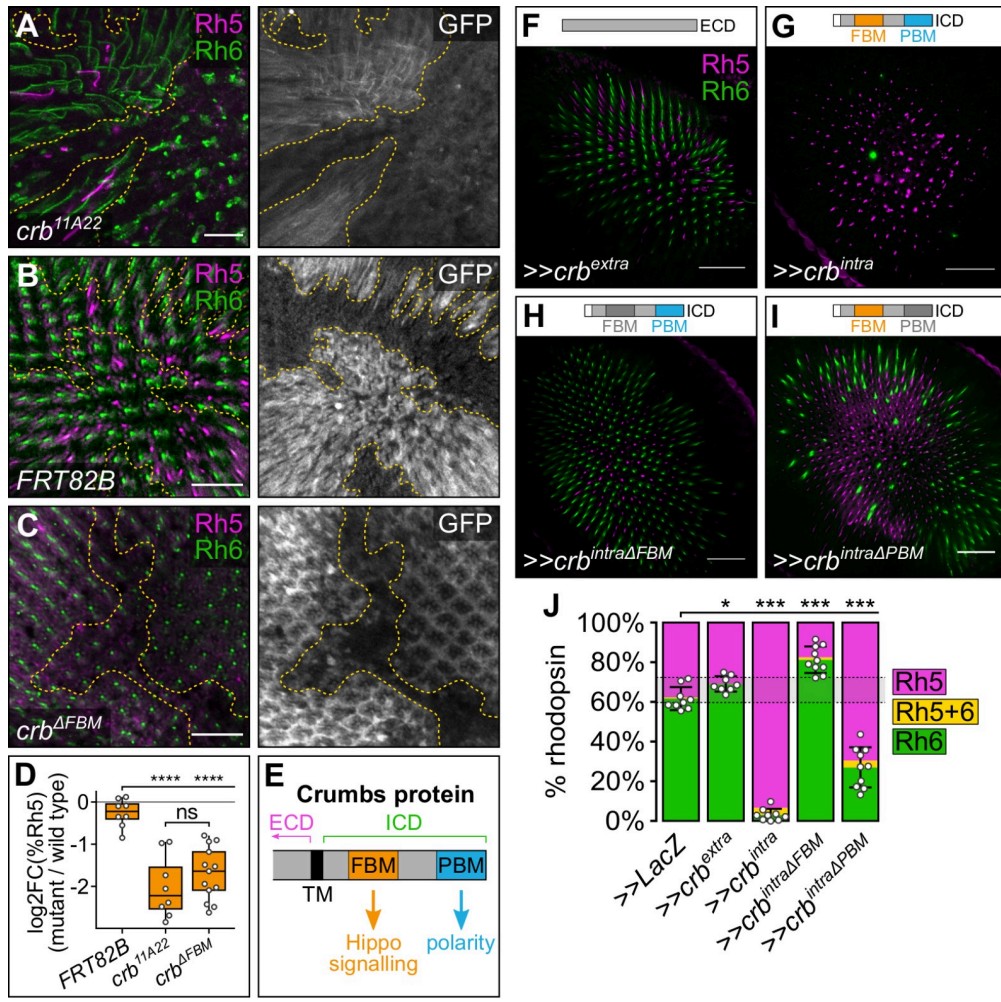

**Fig 5. Crumbs regulates R8 cell fate through its FERM-binding motif.** (**A-C**) Confocal microscope images of adult *D. melanogaster* retinas stained with anti-GFP (grey), anti-Rh5 (magenta) and anti-Rh6 (green) antibodies. GFP-negative clones harboured the following alleles: *crb¹¹ᴬ²²* (**A**), *FRT82B* (negative control) (**B**) or *crbᐃFBM*.*HA* (**C**). Panel (**A**) is a maximum projection as rhodopsins localised to different focal planes in wild type and mutant clones. Scale bars are 20μm. (**D**) Log2 value of the ratio of total % Rh5 (% cells expressing only Rh5 and cells co-expressing Rh5 and Rh6) between mutant and wild type clones from the same tissue. Genotypes were compared with an ANOVA; ns = not significant; **** = p<0.0001. *FRT82B*: n = 8 retinas, 4065 ommatidia; *crb¹¹ᴬ²²*: n = 8, 1394; *crbᐃFBM*.*HA*: n = 10, 3851. (**E**) Schematic illustration of the intracellular domain of the Crb protein. ECD, extracellular domain; ICD, intracellular domain; TM, transmembrane domain; FBM, FERM-binding motif; PBM, PDZ-binding motif. (**F-I**) Confocal microscope images of adult *D. melanogaster* retinas stained with anti-Rh5 (magenta) and anti-Rh6 (green) antibodies. The indicated transgenes were driven by *lGMR-Gal4*. Retinas expressed *crbextra* (**F**), *crbintra* (**G**), *crbintraᐃFBM* (**H**) or *crbintraᐃPBM* (**I**). Schematic illustrations above each retina indicate the transgenes expressed in each experiment; motifs in dark grey indicate mutated motifs in the transgene. Scale bars are 20μm. (**J**) Proportion of R8 cells that express Rh5 (magenta), Rh6 (green), or both (yellow). The error bars represent the standard deviation of total % Rh5 (% cells expressing only Rh5 and cells co-expressing Rh5 and Rh6). Total % Rh5 was compared with two-sided, unpaired t-tests; * = p<0.01, *** = p<0.0001. The shaded grey region between the dotted grey lines indicates wild type Rh5:Rh6 ratio range. *>>LacZ*: n = 9 retinas, 3211 ommatidia; *>>crbextra*: n = 8, 2153; *>>crbintra*: n = 9, 2381; *>>crbintraᐃFBM*: n = 10, 3351; *>>crbintraᐃPBM*: n = 10, 4041.

transmembrane protein composed of a long extracellular domain, a transmembrane domain, and a short intracellular domain. The extracellular domain is essential for Crb apical enrichment and stabilisation, and cell aggregation by mediating Crb-Crb interactions between neighbouring cells [74, 75]. The intracellular domain contains two defined motifs–a juxtamembrane FERM-binding motif (FBM) and a C-terminal PDZ (PSD-95/Dlg/ZO-1)-binding motif

(PBM) (**Fig 5E**). The Crb FBM can bind to FERM domains in proteins such as Ex, to regulate Hippo pathway activity and tissue growth [42, 46–48], while the PBM recruits members of the Crumbs complex (Stardust, Patj and Lin-7) to promote and maintain apicobasal polarity and epithelial integrity [76]. To investigate the role of the Crb FBM in R8 cell fate, we generated clones of $crb^{ΔFBM}$ tissue, an allele with mutations in three key residues in the Crb FBM [42] (**Figs 5C and 5D and S2C**). Notably, in $crb^{ΔFBM}$ R8 cells, rhodopsin localisation extended the whole length of the R8 cell, as in neighbouring wild type R8 cells, indicating that these mutations do not affect rhabdomere morphogenesis, like the $crb^{11A22}$ allele. Furthermore, $crb^{ΔFBM}$ clones showed on average a 3.3-fold decrease in the percentage of **p**R8 cells compared to neighbouring wild type clones (**Figs 5C and S2C**). The magnitude of change in R8 cell ratio was very similar to that observed in $crb^{11A22}$ clones (p = 0.981) (**Fig 5D**) and indicates that Crb normally promotes **p**R8 cell fate through its FBM.

To investigate whether *crb* overexpression is sufficient to perturb R8 cell fate choice, we misexpressed transgenes composed of only the *crb* extracellular ($crb^{extra}$) or intracellular ($crb^{intra}$) domains in all photoreceptor cells (**Fig 5F**, **5G and 5J**). In *lGMR>crb^{extra}* eyes, there was a mild, but statistically significant, decrease in the proportion of **p**R8 cells (approximately 31% **p**R8 cells, p = 0.0074) compared to the *lGMR>LacZ* control (approximately 38% **p**R8 cells), though the average proportion of Rh5-positive **p**R8 cells was still within the wild type range (**Fig 5F and 5J**). Strikingly however, *lGMR>crb^{intra}* eyes had a strong increase in the proportion of **p**R8 cells (around 93% **p**R8 cells, p<0.0001) (**Fig 5G and 5J**). Therefore, in R8 cells, mutating *crb* or misexpressing $crb^{intra}$ led to opposing phenotypes–i.e. a decrease or increase in the proportion of **p**R8 cells, respectively, which phenocopies genetic analysis of the Yki transcription coactivator in R8 cells [15]. Interestingly, this role for Crb in R8 cells contrasts with its role in larval eye and wing imaginal discs, where mutation of *crb* and misexpression of $crb^{intra}$ both cause Yki hyperactivation and tissue overgrowth [42, 46–48]. This suggests that there are important differences in how Crb signals to the Hippo pathway in post-mitotic R8 cells and in growing organs. To confirm that misexpression of $crb^{intra}$ regulates R8 cell fate through the Hippo pathway, we combined this with either depletion of *yki* or misexpression of *kibra*. In both cases, the majority of R8 cells now adopted the **y**R8 cell fate (**S4 Fig**). Furthermore, the observed increase in pR8 cells upon $crb^{intra}$ misexpression corresponded with a decrease in R8 cells expressing *wts-LacZ* (**S5C Fig**). Together, these data indicate that Crb regulates R8 cell fate through the Hippo pathway.

As mutation of the *crb* FBM was sufficient to alter the ratio of R8 subtypes to the same extent as a *crb* null allele, we hypothesised that mutating the FBM in the $crb^{intra}$ transgene ($crb^{intraΔFBM}$) would abolish the effects of $crb^{intra}$ misexpression on R8 cell fate. Indeed, *lGMR>crb^{intraΔFBM}* failed to shift the balance of R8 cells to the **p**R8 fate and, in fact, slightly shifted it to the **y**R8 fate (approximately 18% **p**R8 cells, p<0.0001) (**Fig 5H and 5J**). This suggested that another part of the Crb intracellular domain plays a minor role in R8 cell fate control, with a likely candidate being the PBM, which is essential to Crb's role in photoreceptor morphogenesis [73]. To investigate this, we misexpressed a $crb^{intra}$ transgene with a mutated PBM ($crb^{intraΔPBM}$) in photoreceptor cells. *lGMR>crb^{intraΔPBM}* eyes had an increased proportion of **p**R8 cells compared to *lGMR>LacZ* (around 70% **p**R8 cells, p<0.0001), though not to the same extent as in *lGMR>crb^{intra}* eyes (p<0.0001) (**Fig 5I and 5J**). Collectively, this suggests that while the Crb FBM plays a major role in promoting **p**R8 cell fate, the PBM plays a minor role in promoting **y**R8 cell fate.

## Crumbs regulates R8 cell fate independent of Kibra

In *D. melanogaster* larval imaginal discs, Crb can regulate the Hippo pathway by interacting with different upstream Hippo pathway proteins (**Fig 1C**). Crb directly interacts with Ex, both

recruiting it to the apical membrane to promote activation of the pathway [42, 46, 48, 54, 60, 77, 78] and promoting ubiquitin-mediated degradation of Ex [79, 80]. Crb also represses Kibra by sequestering it at sub-apical regions. In the absence of Crb, Kibra localises at the medial apical cortex and recruits Mer, Sav and Wts to this membrane domain, thus activating the Hippo pathway core cassette [43]. As Ex does not regulate R8 cell fate [62], we hypothesised that Crb might regulate the Hippo pathway in R8 cells by repressing Kibra. To investigate the relationship between Crb and Kibra in R8 cells, we generated both *kibra⁴ crb¹¹ᴬ²²* and *kibra⁴ crbᴬꟳᴮᴹ* double mutant clones. Mutant clones for *kibra⁴*, a null allele [37], had a dramatic expansion (10.2 times higher) in the proportion of **p**R8 cells relative to wild type clones (**S3 Fig**), consistent with published studies [62]. Similarly, both *kibra⁴ crb¹¹ᴬ²²* and *kibra⁴ crbᴬꟳᴮᴹ* clones also had a greatly increased proportion of **p**R8 cells compared to wild type cells from the same tissue (15 and 26 times higher, respectively) (**Figs 6A–6C and S2C**), indicating that Kibra acts downstream of, or in parallel to, Crb in R8 cells.

To investigate this further, we assessed whether Crb regulates Kibra subcellular localisation in R8 cells. To do this, we generated an endogenously tagged Kibra-Venus *D. melanogaster* strain using CRISPR-Cas9 genome editing, which displayed a normal R8 subtype ratio indicating that Kibra function was not compromised (**S7 Fig**). In wild-type adult R8 photoreceptor cells, Kibra-Venus was only weakly expressed and visible throughout the cytoplasm (**Fig 6D**). We predicted that in the absence of *crb*, Kibra-Venus would relocalise from the cytoplasm to the rhabdomere, a potentially analogous membrane domain to the medial apical cortex in larval imaginal discs. However, the subcellular localisation of Kibra-Venus was unaltered in *crbᴬꟳᴮᴹ* clones (**Fig 6E**). Similarly, misexpression of the *crbintra* transgene in photoreceptor cells did not alter Kibra-Venus localisation, nor did misexpression of either the *crbintraΔFBM* or *crbintraΔPBM* transgenes (**Fig 6F–6H**). Collectively, these data suggest that Crb does not obviously regulate the Hippo pathway by controlling Kibra subcellular localisation in R8 cells, as it does in growing wing imaginal discs.

The Crb FBM can interact with other proteins, such as Yurt (Yrt), a FERM domain protein [81]. In photoreceptor cells, Yrt interacts with Crb, and mutation of *yrt* results in expanded stalk membranes, the opposite phenotype to that associated with the *crb¹¹ᴬ²²* null mutant, though this does not result in mislocalisation of either Crb or Kst [81]. To investigate a potential role for Yrt in R8 cells, we depleted *yrt* in all photoreceptors by RNAi and observed an increased proportion of **p**R8 cells (45–62% **p**R8 cells across two RNAi lines, p<0.0001) (**S3 Fig**). This suggests that Yrt is important for R8 cell fate as well as photoreceptor morphogenesis.

## Kibra and Merlin do not regulate the Hippo pathway at the rhabdomere

In larval imaginal wing discs, Kibra recruits Hippo pathway components, notably Mer, Sav, Hpo and Wts, to the medial apical cortex to promote Hippo pathway activation [43]. We predicted that if Kibra regulates the Hippo pathway at the medial apical cortex in R8 cells as it does in larval imaginal discs, Kibra overexpression would cause Mer to accumulate at the rhabdomere. To visualise Mer in R8 cells, we generated an endogenously tagged Mer-Venus *D. melanogaster* strain using CRISPR-Cas9 genome editing, which displayed a normal R8 subtype ratio indicating that Mer function was not compromised (S7 **Fig**). We found that in both control and *kibra*-overexpressing R8 cells, Mer-Venus predominantly localised at the stalk membrane (**Fig 6I and 6J**). This suggests that in R8 cells, the Hippo pathway is not obviously activated at membrane domains that are analogous to the medial apical cortex of wing imaginal disc cells.

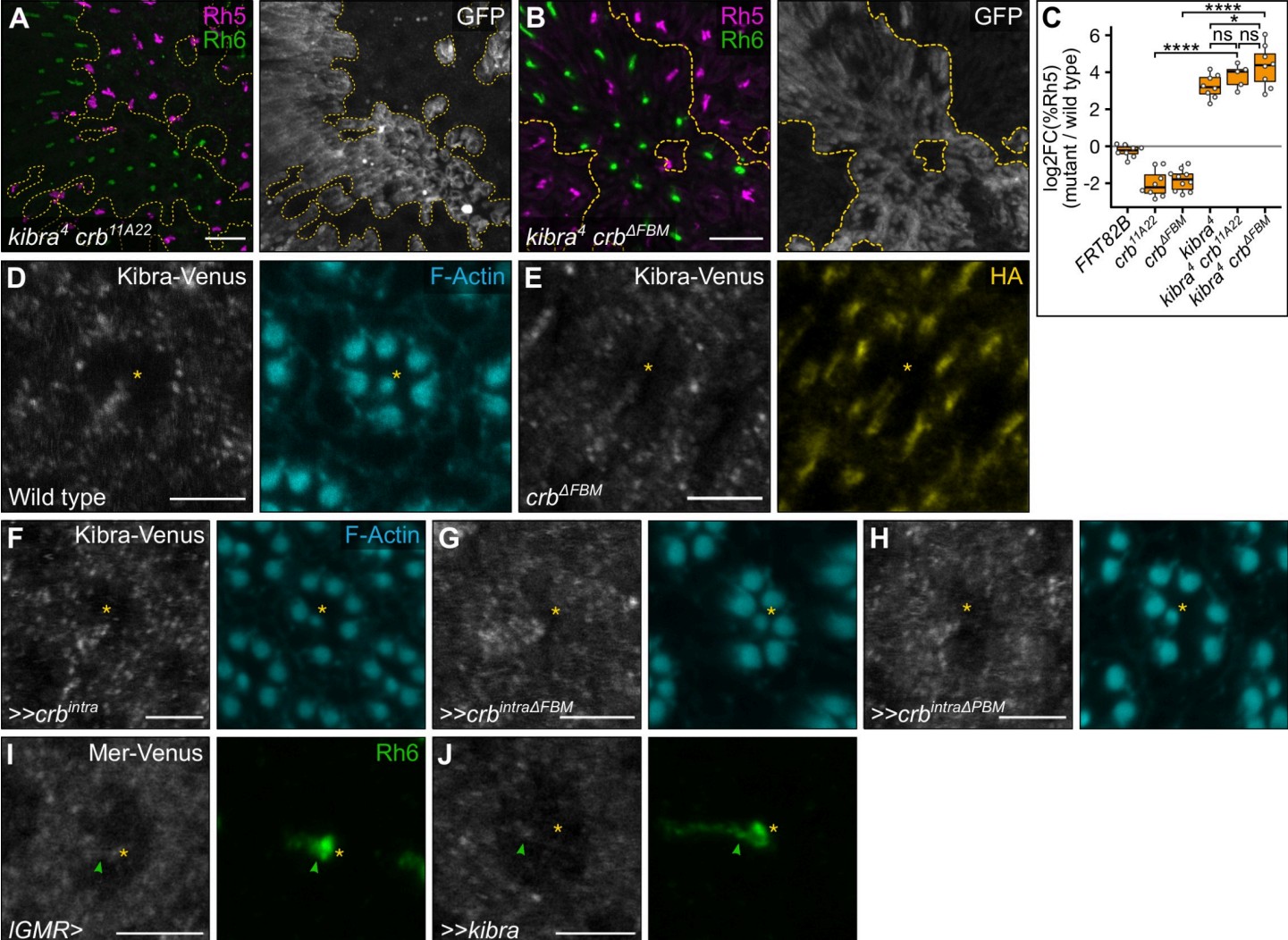

**Fig 6. Crumbs does not affect the subcellular localisation of Kibra in R8 cells.** (**A-B**) Confocal microscope images of adult *D. melanogaster* retinas stained with anti-GFP (grey), anti-Rh5 (magenta) and anti-Rh6 (green) antibodies. GFP-negative clones possessed the following alleles: *kibra⁴ crb¹¹ᴬ²²* (**A**), or *kibra⁴ crbᴬᶠᴮᴹ* (**B**). Panel (**A**) is a maximum projection as rhodopsins localised in different focal planes in wild type and mutant clones. Scale bars are 20μm. (**C**) Log2 value of the ratio of total % Rh5 (% cells expressing only Rh5 and cells co-expressing Rh5 and Rh6) between mutant and wild type clones from the same tissue. Genotypes were compared with an ANOVA; ns = not significant; *** = p<0.001. *FRT82B* (**Fig 4B**): n = 8 retinas, 4065 ommatidia; *crb¹¹ᴬ²²* (**Fig 4A**): n = 8, 1394; *crbᴬᶠᴮᴹ.HA* (**Fig 4C**): n = 10, 3851; *kibra⁴*: n = 8, 2776; *kibra⁴ crb¹¹ᴬ²²*: n = 5, 1174; *kibra⁴ crbᴬᶠᴮᴹ*: n = 8, 2479. (**D-E**) Confocal microscope images of adult *D. melanogaster* retinas stained with anti-GFP (grey) antibody and either anti-HA antibody (yellow) or Phalloidin (F-actin, cyan). GFP-positive clones expressed *kibra-Venus* in wild-type cells (**D**) or cells harbouring the *crbᴬᶠᴮᴹ.HA* allele (**E**). Scale bars are 5μm. (**F-H**) Confocal microscope images of *kibra-Venus* adult *Drosophila* retinas stained with anti-GFP antibody (grey). The indicated transgenes were driven by *lGMR-Gal4*. Retinas expressed *crbⁱⁿᵗʳᵃ* (**F**), *crbⁱⁿᵗʳᵃᴬᶠᴮᴹ* (**G**) or *crbⁱⁿᵗʳᵃᴬᴾᴮᴹ* (**H**). Scale bars are 5μm. (**I-J**) Confocal microscope images of *Mer-Venus* adult *D. melanogaster* retinas stained with anti-GFP (grey) and anti-Rh6 (green) antibodies. The indicated transgenes were driven by *lGMR-Gal4*. Retinas expressed no transgene (**I**) or *kibra* (**J**). Yellow stars indicate the R8 rhabdomere; green arrows indicate the stalk of the R8 cell. Scale bars are 5μm.

## Discussion

The Hippo pathway is a complex signalling network that integrates multiple signals to control organ growth and cell fate decisions, including the binary fate choice of R8 photoreceptors in the *D. melanogaster* eye [14]. The proteins that take part in Hippo pathway signal transduction in organ growth are better understood than those in cell fate. Here, we identify the apical spectrin cytoskeleton proteins α-Spec and Kst, and the apicobasal polarity protein Crb, as important regulators of the R8 cell fate choice. By contrast, neither β-Spec nor Jub, which operate in

the Hippo pathway in tissues such as the imaginal discs and ovary, regulate R8 cell fate. Therefore, we provide new information on R8 cell fate specification and how the Hippo pathway mediates signal transduction in different biological settings.

Interestingly, our study suggests that Crb and the apical spectrin cytoskeleton each transduce signals to the Hippo pathway via distinct modes in organ growth and R8 cell fate specification. While both mutation and overexpression of *crb* in growing wing and eye imaginal discs causes Yki hyperactivity and tissue overgrowth [42, 46–48], loss of *crb* in R8 cells led to a decrease in **p**R8 cells–synonymous with reduced Yki activity–while *crb* overexpression increased **p**R8 cells–a phenotype associated with Yki hyperactivity. In growing larval imaginal discs, Crb has opposing influences on Hippo pathway activity via three mechanisms: (1) it recruits Ex to the sub-apical regions, leading to Hippo pathway activation; (2) it promotes the ubiquitination and degradation of Ex, resulting in suppression of Hippo pathway activity; and (3) it sequesters Kibra at sub-apical regions, limiting activation of the Hippo pathway at the medial apical cortex (**Fig 7A**) [42, 43, 46–48, 79]. In growing larval imaginal discs, Ex's role as an activator of the Hippo pathway must dominate over the other mechanisms, as Crb loss impedes Hippo pathway activity [42, 46–48]. By contrast, in R8 cells, Crb appears to primarily mediate a Hippo-inhibitory signal that promotes Yki activity. This apparent change in Crb signalling to the Hippo pathway could be explained by the low expression of Ex in pupal and adult eyes [82] and the dispensability of *ex* for R8 cell fate [62]. Accordingly, only the Kibra-antagonism function of Crb might operate in R8 cells. Consistent with this, we found that *kibra* loss was completely epistatic to *crb* loss in R8 cells. However, unlike in growing imaginal discs [43], we found no evidence that Crb influences the subcellular localisation of Kibra in R8 cells, although these studies were technically challenging because of the very low expression of Kibra.

Our genetic analysis of *crb* in R8 cells suggests a fourth mechanism by which Crb signals to the Hippo pathway via an unidentified FERM-domain protein that suppresses Hippo pathway activity (**Fig 7B**). Candidate FERM-domain proteins that might regulate the Hippo pathway in conjunction with Crb include Moesin (Moe) and Yurt (Yrt), which can physically interact with Crb [70, 81, 83]. While neither Moe nor Yrt have been directly associated with Hippo signalling, Moe associates with both Crb and Kst [70] and plays a role in photoreceptor morphogenesis [84], while Yrt is a negative regulator of Crb in photoreceptor development [81, 83]. Consistent with Yrt being an inhibitor of Crb, we found that Yrt loss promotes the **p**R8 cell fate, although the exact mechanism by which this occurs requires further investigation. This putative Crb-dependent regulatory mechanism of the Hippo pathway might be R8-specific or also operate more broadly, for example in growing larval imaginal discs. Crb is known to play an important role in photoreceptor morphogenesis during pupal development [67]. Our results suggest that this role of Crb in regulating morphogenesis is distinct from its role in regulating R8 cell fate described here. We showed that mutating only the Crb FBM is sufficient to alter the R8 subtype ratio and this phenotype is present even in the absence of any defects in photoreceptor morphogenesis. We hypothesise that the role of Crb in R8 cell fate is also distinct from its role in regulating cell polarity, as mutating its PBM, which mediates the formation of the Crb polarity complex [76], failed to fully rescue the increased proportion of **y**R8 cells seen upon misexpression of *crb*.

Another important consideration on Crb's role in R8 cell fate is whether it conveys signals that are mediated by its extracellular domain. In larval imaginal discs, Crb engages in homophilic interactions between apical junctions of neighbouring cells and this has been hypothesised to be important for its ability to control Hippo signalling and cell competition [85]. The Crb extracellular domain is also important for stability of the Crb protein and the maintenance of apicobasal polarity [74]. The role of the Crb extracellular domain in R8 cell fate is currently

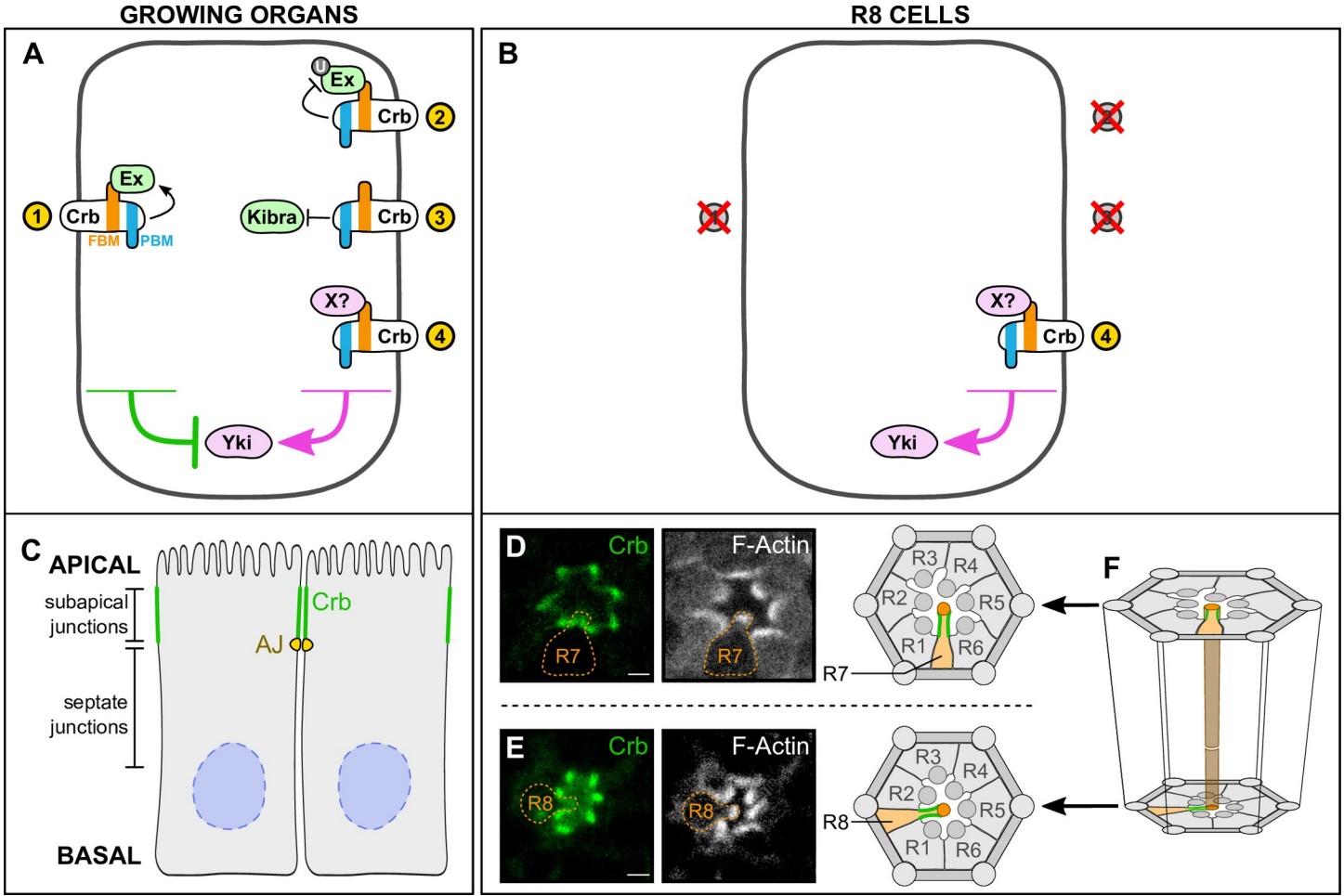

**Fig 7. Model of Crumbs function in growing organs and R8 cells.** (**A-B**) Schematic diagram of the role of Crb in the Hippo pathway in growing organs (**A**) and R8 cells (**B**). Proteins and arrows in magenta promote organ growth or **p**R8 cell fate; proteins and arrows in green suppress organ growth or promote **y**R8 cell fate. Crb, Crumbs; Ex, Expanded; FBM, FERM-binding motif; PBM, PDZ-binding motif; Yki, Yorkie. (**C-F**) Subcellular ocalisation of Crb in epithelial cells and R8 photoreceptor cells. (**C**) Schematic diagram of epithelial cells, with Crb localisation (green) at the sub-apical regions. Adherens junctions (AJ) are depicted in yellow, nuclei in blue. (**D-E**) Confocal microscope images of a *Crb-GFP* pupal ommatidium stained with Phalloidin (F-Actin). R7 and R8 cells are outlined in orange. Scale bars are 2μm. (**F**) Schematic diagram of an ommatidium, showing R7 and R8 planes from (**D**) and (**E**). The brown tube at the centre of the diagram indicates the optic path shared by the rhabdomeres of the R7 and R8 cells. Crb is shown in green; R7 and R8 cells are shown in orange.

unclear although its misexpression did not influence R8 cell fate choice. R8 fate is induced by the neighbouring R7 cells [86–88] and is conveyed by parallel Activin and BMP signalling, however how Hippo pathway activity is influenced in R8 cells has not yet been elucidated [13]. One possibility is that Crb engages in homophilic interactions and thereby signals from R7 to R8 cells, although based on its subcellular localization in these cells, we think this is unlikely. In photoreceptor cells, Crb localises at the membrane of the stalk, the apical subcellular compartment of photoreceptor cells located basally of the rhabdomere [73, 89], analogous to its localisation at the sub-apical regions in imaginal disc epithelial cells [76] (**Fig 7C–7F**). However, while Crb is directly apposed in neighbouring imaginal disc epithelial cells and allows for interactions between Crb extracellular domains of adjacent cells (**Fig 7C**), the stalks of the R7 and R8 cells do not obviously overlap given that the R7 cell is positioned distally and the R8 cell is positioned proximally (**Fig 7C–7F**). Therefore, Crb is unlikely to signal between neighbouring R7 and R8 cells via homophilic interactions between the Crb extracellular domain.

Another apparent difference in Hippo signalling between growing organs and R8 cells relates to the apical spectrin cytoskeleton. As described above, in growing larval imaginal discs, the apical spectrin cytoskeleton has been proposed to regulate the Hippo pathway by two modes: (1) by binding to both Crb and Ex, which recruit the core kinase cassette to sub-apical regions to be activated [54]; and (2) by influencing cytoskeletal tension and, thereby, Jub-dependent suppression of Wts at adherens junctions [53, 57]. Here, we found that depletion of Crb and the apical spectrin cytoskeleton components, α-Spec and Kst, have an opposing impact on R8 cell fate. This suggests that their regulatory roles are decoupled in the context of R8 cell fate, which is particularly surprising since Crb and the apical spectrin cytoskeleton co-ordinately regulate photoreceptor morphogenesis during pupal development [67]. Additionally, we showed that depletion of the apical spectrin cytoskeleton leads to an increase of pSqh, as in the pupal eye [53]. However, we were unable to detect expression of Jub in adult photoreceptors and found no role for Jub in R8 cell fate choice. As such, the mechanism by which the apical spectrin cytoskeleton influences Hippo pathway activity in R8 cells is currently unclear, but seems to be distinct from those that operate during organ growth.

The Hippo pathway is important for cell fate determination in a number of tissues in addition to the R8 cells of the *D. melanogaster* eye, including the posterior follicle cells of the *D. melanogaster* egg chamber [90–93], the periopodial epithelium/disc proper cell fate decision of the *D. melanogaster* larval eye imaginal disc [94] and the inner cell mass/trophectoderm decision in the early mouse blastocyst [95–99]. In posterior follicle cells, as in R8 cells, some Hippo pathway components, such as the Fat branch of the pathway, are not involved in inducing cell fate [90–92]. Interestingly, mutations in *α-Spec* and *β-Spec*, but not in *kst* or *crb*, stimulate Yki activity and proliferation of posterior follicle cells [53–55], indicating that the basolateral spectrin cytoskeleton, rather than the apical spectrin cytoskeleton, regulates the Hippo pathway in these cells. Combined with our results, this suggest that different cells have repurposed different components of the Hippo pathway to control cell fate. Defining the signalling logic employed by the Hippo pathway to control different cell fate choices and tissue growth, should reveal new insights into these biological processes and also how cellular machinery is redeployed in living systems.

## Materials and methods

### *Drosophila melanogaster* genetics

The following *D. melanogaster* stocks were used, many available from the Bloomington *Drosophila* Stock Centre (BDSC), the Vienna *Drosophila* Resource Centre (VDRC), the Kyoto Stock Centre (KSC) and the National Institute of Genetics (Japan) (NIG): lGMR-Gal4 (Claude Desplan), *de-Gal4* [100], *UAS-β-gal RNAi$^{GD}$* (VDRC, #51446), *UAS-yki RNAi$^{KK}$* (VDRC, #104523), *UAS-wts RNAi$^{KK}$* (VDRC, #106174), *UAS-α-Spec RNAi$^{GD}$* (VDRC, #25387), *UAS-α-Spec RNAi$^{TRiP}$* (VDRC, #56932), *UAS-β-Spec RNAi$^{TRiP}$* (BDSC, #38533), *UAS-β-Spec RNAi$^{GD}$* (VDRC, #42053), *UAS-β-Spec RNAi$^{GD}$* (VDRC, #42054), *UAS-kst RNAi$^{TRiP}$* (BDSC, #33933), *UAS-kst RNAi$^{GD}$* (VDRC, #37074), *UAS-jub RNAi$^{GD}$* (VDRC, #38443), *UAS-jub RNAi$^{KK}$* (VDRC, #101993), *UAS-yrt RNAi$^{TRiP1\ (JF03429)}$* (BDSC, #31771), *UAS-yrt RNAi$^{TRiP2\ (HMS01532)}$* (BDSC, #36118), *UAS-kibra RNAi$^{KK}$* (VDRC, #106507), *UAS-mer RNAi* (NIG, #14228R-2), *FRT82B crb$^{11A22}$* [72, 101], *FRT82B crb$^{ΔFBM\ (Y10AP12AE16A)}$.HA$^{[w+\ GMR]}$* [42], *UAS-crb$^{extra}$*, *UAS-crb$^{intra}$* [102], *UAS-crb$^{intraΔFBM\ (Y10AE16A)}$*, *UAS-crb$^{intraΔPBM\ (ΔERLI)}$* [103], *UAS-sqh.EE*, *UAS-rok.CA* [59], *FRT82B kibra$^4$* [37], *Kst-Venus* (KSC, #115285), *Kibra-Venus* and *Mer-Venus* (generated for this study).

*D. melanogaster* were raised at room temperature (22–23˚C) or 18˚C on food made with yeast, glucose, agar and polenta. Animals were fed in excess food availability to ensure that

nutritional availability was not limiting. All experiments were carried out at 25˚C. Males and females were used for all experiments. Mutant clones were generated using the *eyFlp/FRT* system to generate mutant clones in *D. melanogaster* eye [71].

**Generation of *kibra-venus* and *merlin-venus* D. melanogaster strains.** The *kibra-venus* strains and *mer-venus* were generated by CRISPR/Cas9-mediated targeted transgene integration [104–106]. The gene encoding Venus fluorescent protein was inserted immediately in front of the stop codon of the *kibra* gene or *mer* gene so that Kibra-Venus and Mer-Venus were translated as C-terminal fusion proteins. The donor vectors carried approximately 1kb homology arms on either side of a knock-in cassette comprising genes encoding Venus and 3xP3-RFP [107] flanked by loxP sites. The gRNA expression vectors included a 20-bp proto-spacer sequence, which was designed to include the Kibra and Mer stop codons. The donor and gRNA vectors were co-injected into fertilised eggs laid by nos-Cas9 flies [108]. Transformants were selected by eye-specific red fluorescence of the 3xP3-RFP transgene, which were subsequently removed by crossing to hs-Cre.

## Immunostaining and microscopy

Dissections were performed as described in Hsiao, et al. [109]. Briefly, retinas were dissected in PBS and fixed in 4% paraformaldehyde, washed in PBS for one hour and rinsed in PBST (PBS with 0.3% Triton X-100). During this wash, the lamina was removed. Retinas with strong pigment were washed in PBST for 4–5 days, with the media refreshed once a day, to remove the pigment. Retinas were blocked in blocking solution (5% NGS in PBST) and incubated in primary antibody overnight. Following a one hour wash in PBST, retinas were incubated overnight in secondary antibody. Tissues were mounted in VectaShield Mounting Medium (Vector Laboratories, H-1000) or 90% glycerol on bridge slides [110]. The following primary antibodies were used: mouse anti-Rh5 (1:200, Claude Desplan), rabbit anti-Rh6 (1:1000, Claude Desplan), chicken anti-GFP (1:1000, Abcam, ab13970), chicken anti-β-gal (1:1000, GeneTex, GTX77365), rat anti-Ecad (1:50, Developmental Studies Hybridoma Bank (DSHB), DCAD2), rat anti-HA (1:100, Santa Cruz, 3F10), mouse anti-α-Spec (1:100, DSHB, 3A9), mouse anti-pSqh (1:50, Cell Signalling, 3671S), rat anti-Ci (1:10, DSHB, 2A1). Secondary antibodies conjugated to Alexa405, Alexa488, Alexa555 and Alexa647 (Life Technologies and Invitrogen) were used at a concentration of 1:500. DAPI (1:500, Sigma-Aldrich, D9542) and Phalloidin-TRITC (1:200–500, Sigma-Aldrich, P1951) staining was completed before mounting. Images were collected on a Nikon C2 or Olympus FV3000 confocal microscope, or an Olympus FVMPE-RS multiphoton microscope.

## Image analysis and statistics

Images were analysed using FIJI/ImageJ (https://imagej.net/Fiji). To calculate pSqh antibody staining intensity, figures were cropped so only ommatidia with R8 cells were visible. The mean grey area was calculated from each figure using the Measurement tool. All statistical analyses were completed in RStudio using the stats package. All graphs were generated in RStudio using the ggplot [111] and ggbeeswarm [112] packages. The number of R8 cells that expressed Rh5, Rh6 or both, were counted using the FIJI Cell Counter plugin. Retinas were scored only if there were more than 100 ommatidia in a single focal plane. Statistical comparisons between ratios of R8 subtypes was calculated from the total number of Rh5-positive cells (cells expressing only Rh5 and cells expressing both Rh5 and Rh6) using a two-tailed, unpaired t-test, with the following symbols used for p-value cut-offs: $^{***} < 0.0001$, $^{**} < 0.001$, $^{*} < 0.01$, ns $> 0.01$. Error bars represent the standard deviation. Statistical comparison of the ratio of R8 subtypes in clonal tissues was calculated using an ANOVA and multiple comparisons between

genotypes calculated using Tukey's Honest Significant Difference test, with the following symbols used for p-value cut-offs: **** < 0.0001, *** < 0.001, ** < 0.01, * < 0.05, ns > 0.05.

## Supporting information

**S1 Fig. The impact of RNAi-mediated depletion of Spectrins and *ajuba* in R8 cells.** (**A-E**) Confocal microscope images of adult *D. melanogaster* retinas stained with anti-Rh5 (magenta) and anti-Rh6 (green) antibodies. The indicated RNAi lines were driven by *lGMR-Gal4*. Retinas expressed *α-Spec RNAi$^{TRiP}$* (**A**), *β-Spec RNAi$^{GD\ \#42053}$* (**B**), *β-Spec RNAi$^{GD\ \#42054}$* (**C**), *kst RNAi$^{GD}$* (**D**) or *jub RNAi$^{KK}$* (**E**). Scale bars are 50μm.
(TIF)

**S2 Fig. Validation of Ajuba-GFP and *ajuba* RNAi.** (**A-B'**) Confocal microscope images of third instar larval *D. melanogaster* imaginal wing discs from *Jub-GFP* animals that also expressed one of two *jub RNAi* lines in the posterior compartment, under control of *en-Gal4*. Tissues were stained with anti-Ci (magenta) antibody to mark the anterior compartment (left) and DAPI (cyan) to mark nuclei. The genotypes for each tissue are: *en>jub RNAi$^{KK}$; Jub-GFP* (**A**) and *en>jub RNAi$^{GD}$; Jub-GFP* (**B-B'**). Scale bars are 100μm in **A** and **B** and 10μm in **B'**.
(TIF)

**S3 Fig. Investigation of Crumbs, Kibra and Yurt in R8 cells.** (**A**) Confocal microscope images of adult *D. melanogaster* retinas stained with anti-GFP (grey) and anti-Rh6 (green) antibodies and stained with Phalloidin (cyan) to visualise the rhabdomeres. GFP-negative clones were mutant for *crb$^{11A22}$*. Arrowheads indicate a wild type ommatidium (magenta) and a mutant ommatidium (white). Scale bar is 50μm. (**B**) Confocal microscope images of adult *D. melanogaster* retinas stained with anti-GFP (grey), anti-Rh5 (magenta) and anti-Rh6 (green) antibodies. GFP-negative clones were mutant for *kibra$^4$*. Scale bar is 50μm. (**C**) Proportion of R8 cells in wild type ('+') or mutant ('–') clones that express Rh5 (magenta), Rh6 (green), or both (yellow). Grey lines connect wild type and mutant clones from the same retina. The error bars represent the standard deviation of total % Rh5 (% Rh5 + % Rh5+Rh6). Total % Rh5 was compared with two-sided, unpaired t-tests; ns = not significant, *** = p<0.0001. The shaded grey region between the dotted grey lines indicates wild type Rh5:Rh6 ratio range. *FRT82B*: n = 8 retinas, 4065 ommatidia; *crb$^{11A22}$*: n = 8, 1394; *crb$^{AFBM}$.HA*: n = 10, 3851; *kibra$^4$*: n = 8, 2776; *kibra$^4$ crb$^{11A22}$*: n = 5, 1174; and *kibra$^4$ crb$^{AFBM}$.HA*: n = 8, 2479. (**D-E**) Confocal microscope images of adult *Drosophila* retinas stained with anti-Rh5 (magenta) and anti-Rh6 (green) antibodies. Retinas expressed either *UAS-yrt RNAi$^{TRiP1\ (JF03429)}$* (**D**) or *UAS-yrt RNAi$^{TRiP2\ (HMS01532)}$* (**E**). Scale bars are 20μm. (**F**) Proportion of R8 cells that express Rh5 (magenta), Rh6 (green), or both (yellow). The error bars represent the standard deviation of total % Rh5 (% Rh5 + % Rh5+Rh6). Total % Rh5 was compared with two-sided, unpaired t-tests; *** = p<0.0001. The shaded grey region between the dotted grey lines indicates wild type Rh5:Rh6 ratio range. *β-gal RNAi* (Fig 1D): n = 9 retinas, 3976 ommatidia; *UAS-yrt RNAi$^{TRiP1\ (JF03429)}$*: n = 7, 1740; *UAS-yrt RNAi$^{TRiP2\ (HMS01532)}$*: n = 7, 1224.
(TIF)

**S4 Fig. The apical spectrin cytoskeleton and Crumbs regulate R8 cell fate upstream of Kibra and Yorkie.** (**A-L**) Confocal microscope images of adult *Drosophila* retinas stained with anti-Rh5 (magenta), anti-Rh6 (green) and anti-β-gal (grey; Wts-LacZ) antibodies. Retinas expressed either *UAS-α-Spec RNAi* (**A-D**), *UAS-kst RNAi* (**E-H**), or *UAS-crb$^{intra}$* (**I-L**) in conjunction with either *UAS-yki RNAi* (**A, E, I**), *UAS-β-gal RNAi* (**B, F, J**), *UAS-kibra* (**C, G, K**), or *UAS-LacZ* (**D, H, L**). Scale bars are 20μm. (**M**) Proportion of R8 cells that express Rh5

(magenta), Rh6 (green), or both (yellow). The error bars represent the standard deviation of total % Rh5 (% Rh5 + % Rh5+Rh6). Total % Rh5 was compared with two-sided, unpaired t-tests; * = p<0.01, ** = p<0.001, *** = p<0.0001. The shaded grey region between the dotted grey lines indicates wild type Rh5:Rh6 ratio range. *lGMR>α-Spec RNAi>β-gal RNAi*: n = 3 retinas, 342 ommatidia; *lGMR>α-Spec RNAi>yki RNAi*: n = 3, 725; *lGMR>α-Spec RNAi>LacZ*: n = 3, 436; *lGMR>α-Spec RNAi>kibra*: n = 3, 442; *lGMR>kst RNAi>β-gal RNAi*: n = 3, 462; *lGMR>kst RNAi>yki RNAi*: n = 4, 589; *lGMR>kst RNAi>LacZ*: n = 3, 908; *lGMR>kst RNAi>kibra*: n = 4, 791; *lGMR>crb^{intra}>β-gal RNAi*: n = 3, 499; *lGMR>crb^{intra}>yki RNAi*: n = 3, 835; *lGMR>crb^{intra}>LacZ*: n = 3, 628; *lGMR>crb^{intra}>kibra*: n = 3, 885.
(TIF)

**S5 Fig. In R8 cells, *warts-LacZ* is expressed only in the yR8 subtype upon modulation of the apical spectrin cytoskeleton or Crumbs.** (**A-C**) Confocal microscope images of adult *Drosophila* retinas stained with anti-Rh5 (magenta), anti-Rh6 (green) and anti-β-gal (grey/orange; *wts-LacZ*) antibodies. Retinas expressed either *UAS-α-Spec RNAi* (**A**), *UAS-kst RNAi* (**B**), or *UAS-crb^{intra}* (**C**). Blue circles indicate examples of *wts-LacZ*-positive cells which were also Rh6-positive. Scale bars are 20μm. Note that *wts-LacZ* is also expressed in the interommatidial cells, though at lower levels than in R8 cells; this is particularly evident in the retinas expressing *crb^{intra}*.
(TIF)

**S6 Fig. Depletion of the apical spectrin cytoskeleton does not affect Spaghetti squash localization in the adult retina.** (**A-C'**) Confocal microscope images of adult *Drosophila* retinas stained with anti-Rh5 (magenta), and anti-Rh6 (green) antibodies showing Sqh-GFP localisation. Retinas expressed *UAS-α-Spec RNAi* (**A**), *UAS-kst RNAi* (**B**) or *Sqh-GFP*, alone (**C**). Scale bars are 10μm. (**D-E**) Confocal microscope images of adult *Drosophila* retinas stained with anti-Rh5 (magenta), and anti-Rh6 (green) antibodies. Retinas expressed *UAS-sqh.EE* (**D**), and *UAS-rok.CA* (**E**). Scale bars are 20μm. (**F**) Proportion of R8 cells that express Rh5 (magenta), Rh6 (green), or both (yellow). The error bars represent the standard deviation of total % Rh5 (% Rh5 + % Rh5+Rh6). Total % Rh5 was compared with two-sided, unpaired t-tests; ns = not significant, *** = p<0.0001. The shaded grey region between the dotted grey lines indicates wild type Rh5:Rh6 ratio range. *>>LacZ*: n = 9 retinas, 3211 ommatidia; *>>sqh.EE*: n = 7, 1355; *>>rok.CA*: n = 6, 1314.
(TIF)

**S7 Fig. Validation of *Kibra-Venus* and *Merlin-Venus D. melanogaster* strains.** (**A-B**) Third instar larval wing (**A**) or eye (**B**) imaginal discs from *D. melanogaster* strains expressing endogenously tagged Kibra-Venus (**A**) or Mer-Venus (**B**). In (**A**), tissues were stained with anti-Ci antibody to mark the anterior (wild type) half of the wing disc, while in (**B**) RFP marks the dorsal (RNAi knockdown) half of the eye disc (magenta). DAPI (grey) marks nuclei. Genotypes are *en-Gal4 / kibra RNAi; kibra-Venus / +* (**A**) and *Mer-Venus / +; de-Gal4, UAS-RFP / UAS-mer RNAi* (**B**). In all images, anterior is towards the left and dorsal is towards the top. Scale bars are 100μm. Yellow arrowheads indicate a compartment boundary. (**C-D"**) Confocal microscope images of adult *Drosophila* retinas stained with anti-Rh5 (magenta) and anti-Rh6 (green) antibodies. Genotypes are *kibra-Venus* (**C-C"**) and *mer-Venus* (**D-D"**). Scale bars are 20μm. (**E**) Proportion of R8 cells that express Rh5 (magenta), Rh6 (green), or both (yellow). The error bars represent the standard deviation of total % Rh5 (% Rh5 + % Rh5+Rh6). Total % Rh5 was compared with two-sided, unpaired t-tests; ns = not significant. The shaded grey region between the dotted grey lines indicates wild type Rh5:Rh6 ratio range. *kibra-Venus*: n = 3 retinas, 864 ommatidia; *mer-Venus*: n = 3, 720.
(TIF)

## Acknowledgments

We thank members of the Harvey lab for discussions. We thank C. Desplan, R. Johnston, H. Richardson, K. Irvine, the Bloomington *Drosophila* Stock Center, the Vienna *Drosophila* RNAi Center, the Kyoto Stock Center, the National Institute of Genetics (Japan), the Australian *Drosophila* Research Support Facility (www.ozdros.com), and the Developmental Studies Hybridoma Bank for *D. melanogaster* stocks and antibodies. We acknowledge the Peter Mac Centre for Advanced Histology and Microscopy and support to them from the Peter MacCallum Cancer Foundation and the Australian Cancer Research Foundation.

## Author Contributions

**Conceptualization:** Jonathan M. Pojer, Kieran F. Harvey.

**Formal analysis:** Jonathan M. Pojer, Abdul Jabbar Saiful Hilmi.

**Funding acquisition:** Kieran F. Harvey.

**Investigation:** Jonathan M. Pojer, Abdul Jabbar Saiful Hilmi.

**Project administration:** Kieran F. Harvey.

**Resources:** Shu Kondo.

**Supervision:** Kieran F. Harvey.

**Writing – original draft:** Jonathan M. Pojer, Kieran F. Harvey.

**Writing – review & editing:** Jonathan M. Pojer, Kieran F. Harvey.

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
