## [Decision Letter · Decision Letter 0]

2 Nov 2020

Dear Kieran,

Thank you very much for submitting your Research Article entitled 'Crumbs and the Apical Spectrin Cytoskeleton Regulate R8 Cell Fate in the Drosophila eye' to PLOS Genetics. Your manuscript was fully evaluated at the editorial level and by three independent peer reviewers. As you will see, the reviewers were of different opinions and you will have to satisfy Review  #2 who has the most concerns  about the paper.

These substantial concerns about the current manuscript do now allow us to accept the manuscript. However, we would be willing to review a much-revised version that will  be sent back  to reviewers #1 and #2 (#3 was quite positive already). We cannot, of course, promise publication at that time.

If you decide to revise the manuscript for further consideration at PLOS Genetics, please aim to resubmit within the next 60 days, unless it will take extra time to address the concerns of the reviewers, in which case we would appreciate an expected resubmission date by email to plosgenetics@plos.org.

[LINK]

We are sorry that we cannot be more positive about your manuscript at this stage. Please do not hesitate to contact us if you have any concerns or questions.

Yours sincerely,

Claude

Claude Desplan

Associate Editor

PLOS Genetics

Gregory P. Copenhaver

Editor-in-Chief

PLOS Genetics

Reviewer's Responses to Questions

**Comments to the Authors:**

Reviewer #1: In the present manuscript, Pojer et al explore the role of several upstream Hippo pathway components in R8 photoreceptor specification in the Drosophila retina. They find that, contrary to expectation based on past results in imaginal disc growth, the apical polarity determinants Crb and components of the Spectrin cytoskeleton function in an antagonistic manner in R8 specification. Furthermore, although apical Spectrin depletion leads to elevated pMyoII levels, the effects on R8 specification do not appear to be mediated bu Ajuba as in wing discs. Finally, Crb appears to promote the Yki-dependent “pale” R8 fate, possibly via an unknown FERM domain protein. The manuscript provided new insights into R8 fate specification and illustrate the fact that Hippo pathway upstream wiring is highly context-dependent. The manuscript should be of interest to the field, however, there are several aspects that can easily be improved. I would therefore support publication in PLoS Genetics, provided the points below are addressed.

1. The main shortcoming of the ms is that the Crb and Spectrin manipulations are not directly linked to Yki activity. This can be addressed as follows:

- Look at melted-LacZ and wts-LacZ expression upon loss of spectrin and crb, and upon overexpression of Crb-intra.

- Deplete yki in the context of Crb-intra overexpression and spectrin depletion, and measure R8 Rhosopsin expression.

2. The authors show that pMyoII levels are elevated upon apical Spectrin depletion, yet Ajuba doesn’t appear to be involved as in wing discs. They should explore this further:

- Are the Ajuba RNAi lines they are using really working? Either show this by depletion in the wing disc or make Ajuba mutant clones in the retina.

- Does SqhEE overexpression affect R8 fate, which would suggest that a Jub-independent but MyoII-dependent mechanism downstream of Spectrin is at play here.

3. To try and identify the putative FERM domain protein involved in R8 specification together with Crb, the authors should try to deplete the main candidates Moesin and Yurt and look at Rhodopsin expression.

4. Is Spectrin localisation changed in Crb-mutant clones? It would be interesting to compare the null with the delta-FBM allele.

Minor points:

1. Line 137: The authors state: “Depletion of α-Spec (40-60% pR8 cells across two RNAi lines, p=0.026,<0.0001)” but in Figure 2 one of the α-Spec RNAi lines is marked as not significant.

2. Figure 4A-C: it looks like pSqh staining is reduced in IOCs upon Spectrin depletion. The authors should confirm whether this is the case and comment on this in the text.

3. The Kibra-Venus/Mer-Venus transgene should be described in this manuscript.

Reviewer #2: This is an interesting paper that sets out to study the relationship between the apical protein Crb, some of the proteins with which it has been associated, and the Hippo pathway in cell fate decision, using the fly eye as a model system.

The eye consists of 2 predominant subtypes of ommatidia. Approximately 30% express Rh3 in R7p and Rh5 in the paired R8p, while 70% express Rh4 in R7y and Rh6 in the paired R8y. Choice in Rhodopsin expression is stochastic and relies on a bistable feedback loop that involves components of the Hippo pathway. R8y fate is achieved through activation of Warts, which represses Yorkie activity. Inactivation of Wts promotes the R8p fate.

Crumbs is a known regulator of the Hippo pathway in growth control, but whether and how it might regulate cell fate decision in a cell type like the R8 photoreceptor, has not been explored in detail.

Overall, the paper is not easy to read, and this is largely due to the fact it presents many observations that are not always well integrated in a working model. While this could be addressed by working on the text, coming up with a clearer account of what might be going on will also require more experiments.

Panels appeared to be missing from Figure 6.

The key findings reported here are that:

i) Crb is required during R8p / R8y cell fate decision. Loss of crb function leads to a reduction in R8p cells and increase in R8y. Conversely, overexpression of the Crb intracellular domain, which is known to harbour Crb signalling activity with respect to the Hippo pathway, greatly promotes the R8p fate. This is new and very interesting as it connects a known epithelial polarity determinant to cell fate decision.

ii) The Hippo pathway activator Kibra is a good candidate effector for Crb function in R8 fate decision. It has been proposed that Crb can inhibit the Hippo pathway by sequestering Kibra. Therefore, one might expect that loss of Crb would lead to activation of the Hippo pathway via kibra release. Consistent with this model, Kibra loss of function promotes the R8p fate.

iii) The spectrin cytoskeleton, and in particular the apical portion of it, which is known to interact with Crb, is part of the machinery that regulates R8 cell fate decision.

Collectively, these finding clearly point to a function for Crb and known interactors in R8 cell fate decision. However, the apparent contradiction between the Crb and Spectrin phenotypes and relative lack of a clear understanding of how exactly Kibra fits into this picture, makes it difficult go beyond the notion that things are complex and not do not behave as expected. So while the paper contains new interesting observations, collectively, no clear model comes out of the work. This could probably be addressed by further probing the links between Crb and Kibra and try for example to rescue the Crb loss of function by manipulating Kibra? Asking whether the effect of Crb overexpression on R8 fate decision dependents on Kibra? Similar experiments could be envisaged with respect to Spectrins. This would help to flesh out the functional connection between Kibra, Spectrin and Crb in R8.

I also think that the section related to P-Sqh might be problematic. P-Sqh antibodies are notoriously tricky to use and antibody penetration in the retina is not always good. MyoII has been shown to accumulate at the apical membrane in photoreceptors. The staining presented in Fig4 shows signal in the interommatidial cells, which becomes more diffused in kst and a-spec RNAI retina. No info is provided as to how, the intensity signal was quantified especially in light of the statement that ‘the apical spectrin cytoskeleton regulates phosphorylation of Sqh in R8 cell’– pp8.

I think that perhaps using the SqhGFP (or ZipGFP) fly strain might be more suitable, as it would bypass any technical limitations associated with the P-Sqh Ab. These GFP lines are very good proxies of “active MyoII”.

- The authors should discuss the possibility that by affecting photoreceptor morphogenesis, the crb lof might lead to failure in R7-R8 communication. This might well explain a requirement for the PDM domain.

- Evaluating properly the work on Kibra and Mer requires one to have access to how the Venus transgenes were made. ‘To be described elsewhere’ prevents proper evaluation of this part of the work and is problematic. Are they functional transgenes? do they come with a phenotype? are they expressed at levels that are comparable to the endogenous, none tagged proteins, etc.

- It would be good to have an idea of how reliable the RNAi lines that are used in this study are, especially when the results are negative.

- Please note that the sub-apical region (stalk in the photoreceptor) is not usually referred to as a 'junction'.

Reviewer #3: In this manuscript, Pojer et al investigate the role of the Hippo pathway in the control of fate choice in the R8 photoreceptors. The authors implicate Crumbs (Crb) and apical spectrin cytoskeleton regulators α-Spectrin (α-Spec) and Karst (Kst) in R8 cell fate determination. α-Spec and Kst are necessary for the specification of Rh6-expressing yR8 cells and knockdown of either gene resulted in a greater proportion of Rh5-positive pR8 cells in the adult retina. α-Spec and Kst knockdown also reduced phosphorylated Spaghetti squash (Sqh), but this did not alter warts activity through Ajuba activation. Alternatively, Crb is required for pR8 cell specification and mutations in the Crb and the FERM-binding domain alone significantly increase the proportion of Rh6-positive yR8 cells. This activity is independent of its role as a Kibra inhibitor. The authors conclude that only a subset of Hippo pathway proteins regulate the R8 binary cell fate decision. This study illustrates how the Hippo pathway mediates signal transduction in distinct biological settings and therefore it makes an important contribution to the Hippo field. Manuscript is well written, and the data are convincing.

Comments:

- Crb and spectrins were identified in a genetic screen which the authors plan to describe elsewhere. I suggest including it here, at least, as a summary table that shows how many genes were tested and how many positives were isolated. However, I do not consider this absolutely necessary and leave it at their discretion.

- The authors use, at least, two RNAi lines to confirm the phenotypes, which is appropriate. Is it possible to confirm the efficiency of depletion by IF or RT-PCR/western blot? This is particularly important to substantiate the authors’ conclusions based on the lack of the phenotype in RNAi experiments (line 140, for example).

- Line 230 “Crb promotes yR8 cellfate” should be changed to “Crb promotes pR8 cellfate”.

**Have all data underlying the figures and results presented in the manuscript been provided?**

Reviewer #1: Yes

Reviewer #2: **No: **missing info on 2 transgenes generated by the authors and used in the study.

Reviewer #3: Yes

PLOS authors have the option to publish the peer review history of their article (what does this mean?). If published, this will include your full peer review and any attached files.

Reviewer #1: No

Reviewer #2: No

Reviewer #3: No

---

## [Decision Letter · Decision Letter 1]

11 May 2021

Dear Kieran,

We are pleased to inform you that your manuscript entitled "Crumbs and the Apical Spectrin Cytoskeleton Regulate R8 Cell Fate in the Drosophila eye" has been editorially accepted for publication in PLOS Genetics. Congratulations!

Cordially,

Claude

Claude Desplan

Associate Editor

PLOS Genetics

Gregory P. Copenhaver

Editor-in-Chief

PLOS Genetics

Comments from the reviewers (if applicable):

Reviewer's Responses to Questions

**Comments to the Authors:**

Reviewer #1: The authors addressed my comments and I am happy to recommend publication.

Reviewer #2: I think the revised version is now suitable for publication.

**Have all data underlying the figures and results presented in the manuscript been provided?**

Reviewer #1: Yes

Reviewer #2: Yes

PLOS authors have the option to publish the peer review history of their article (what does this mean?). If published, this will include your full peer review and any attached files.

Reviewer #1: No

Reviewer #2: No

**Data Deposition**

http://datadryad.org/submit?journalID=pgenetics&manu=PGENETICS-D-20-01520R1

**Press Queries**

---

## [Editor Report · Acceptance letter]

2 Jun 2021

PGENETICS-D-20-01520R1 

Crumbs and the Apical Spectrin Cytoskeleton Regulate R8 Cell Fate in the Drosophila eye 

Dear Dr Harvey, 

We are pleased to inform you that your manuscript entitled "Crumbs and the Apical Spectrin Cytoskeleton Regulate R8 Cell Fate in the Drosophila eye" has been formally accepted for publication in PLOS Genetics! Your manuscript is now with our production department and you will be notified of the publication date in due course.

With kind regards,

Olena Szabo

PLOS Genetics

On behalf of:
